# Knowledge translation strategies for policy and action focused on sexual, reproductive, maternal, newborn, child and adolescent health and well-being: a rapid scoping review

Janet A Curran [1,2] Allyson J Gallant [3] Helen Wong,[3] Hwayeon Danielle Shin,[1] Robin Urquhart,[4] Julia Kontak [5] Lori Wozney,[6] Leah Boulos,[5] Zulfiqar Bhutta [7] Etienne V Langlois[8]

For numbered affiliations see end of article.

**Correspondence to**
Dr Janet A Curran;
jacurran@dal.ca

## ABSTRACT

**Objective** The aim of this study was to identify knowledge translation (KT) strategies aimed at improving sexual, reproductive, maternal, newborn, child and adolescent health (SRMNCAH) and well-being.

**Design** Rapid scoping review.

**Search strategy** A comprehensive and peer-reviewed search strategy was developed and applied to four electronic databases: MEDLINE ALL, Embase, CINAHL and Web of Science. Additional searches of grey literature were conducted to identify KT strategies aimed at supporting SRMNCAH. KT strategies and policies published in English from January 2000 to May 2020 onwards were eligible for inclusion.

**Results** Only 4% of included 90 studies were conducted in low-income countries with the majority (52%) conducted in high-income countries. Studies primarily focused on maternal newborn or child health and well-being. *Education* (81%), including staff workshops and education modules, was the most commonly identified intervention component from the KT interventions. Low-income and middle-income countries were more likely to include civil society organisations, government and policymakers as stakeholders compared with high-income countries. Reported barriers to KT strategies included limited resources and time constraints, while enablers included stakeholder involvement throughout the KT process.

**Conclusion** We identified a number of gaps among KT strategies for SRMNCAH policy and action, including limited focus on adolescent, sexual and reproductive health and rights and SRMNCAH financing strategies. There is a need to support stakeholder engagement in KT interventions across the continuum of SRMNCAH services. Researchers and policymakers should consider enhancing efforts to work with multisectoral stakeholders to implement future KT strategies and policies to address SRMNCAH priorities.

**Registration** The rapid scoping review protocol was registered on Open Science Framework on 16 June 2020 (https://osf.io/xpf2k).

### Strengths and limitations of this study

► Our review identified a broad range of knowledge translation (KT) strategies and policies aimed at improving sexual, reproductive, maternal, newborn, child and adolescent health and well-being published since January 2020.

► KT strategies and policies were mapped to the Behaviour Change Wheel to identify and describe intervention and policy elements included in the strategies.

► The terminology around KT varies across countries and health areas.

► As a rapid scoping review, full text and grey literature review and data extraction were carried out by a single reviewer and verified by a second reviewer.

► Although we conducted a systematic search of the grey literature it is possible our findings were impacted by our strict inclusion criteria and potential publication bias

## INTRODUCTION

Progress towards sexual, reproductive, maternal, newborn, child and adolescent health (SRMNCAH) has been highly inequitable to date.[1 2] With the current COVID-19 pandemic, there have been substantive and unprecedented disruptions in essential SRMNCAH services,[3] including emerging data on increased maternal mortality, stillbirth rates, ruptured ectopic pregnancies, unintended pregnancies, maternal depression and limited access to contraceptives.[4 5] The greatest disruptions to essential healthcare services are witnessed in low-income countries.[6] As such, special efforts are needed to support evidence-based interventions to prevent further harm, reduce

preventable deaths and morbidity, and promote equitable distribution of essential interventions for SRMNCAH.

There is a lack of equitable distribution of human resources and essential policy adoptions for SRMNCAH in most countries.[7] COVID-19 has also brought to the forefront the need to develop and implement multisectoral interventions using a whole-of-government approach.[8] Improving SRMNCAH across countries of different income levels will require collective action in terms of generating, sharing, brokering and implementing new knowledge through cross-sectoral and interdisciplinary initiatives.[9] The Partnership for Maternal, Newborn and Child Health (PMNCH), hosted by the WHO, is a global partnership designed to address these SRMNCAH inequities, and improve the health and well-being of all women, children and adolescents.[10]

Interventions shown to be effective by scientific endeavours require efforts to integrate evidence into policy and action. Knowledge translation (KT) is 'a dynamic and iterative process that includes synthesis, dissemination, exchange and ethically sound application of knowledge to improve the health, provide more effective health services and products and strengthen the healthcare system'.[11] KT interventions can support this process by facilitating the uptake of evidence into policy and practice targeting change at the professional, institutional or policy level. There has been a growing number of KT interventions, as well as frameworks, theories and models to guide the selection of KT interventions.[12–14] However, the range of KT strategies related to SRMNCAH improvements remains unknown.

No review to date has explored the range of KT interventions utilised at the level of health system, policy or practice specifically addressing the continuum of SRMNCAH. In light of the global call for action to sustain SRMNCAH, it is critical to understand the implementation of KT strategies that promote evidence-based policy and practice for SRMNCAH.

### Objectives

The aim of this rapid scoping review was to identify existing literature related to KT strategies that promote the uptake of evidence into policy and action focused on improving SRMNCAH and well-being. To achieve this aim, four questions were addressed:

1. What are the common KT strategies and activities used to promote the use of evidence to inform policy and action to improve SRMNCAH and well-being?
2. How are stakeholders involved in designing or implementing these KT strategies and activities?
3. What are the commonly reported outcomes of KT strategies and activities to promote the use of evidence in SRMNCAH and well-being?
4. What are the commonly reported barriers and enablers for using KT strategies and activities to promote the use of evidence in SRMNCAH and well-being?

### METHODS

This rapid scoping review follows the methodological guidance developed by the Joanna Briggs Institute and is reported according to the Preferred Reporting Items for Systematic Reviews and Meta-Analyses-Extension for Scoping Reviews (PRISMA-ScR).[15 16] The rapid scoping review protocol was registered on Open Science Framework on 16 June 2020 (https://osf.io/xpf2k).

### Inclusion and exclusion criteria

All study designs were eligible for inclusion in the review. Studies were excluded if they focused on basic science or clinical management of women, newborn, child or adolescent aspects of health or well-being. Systematic reviews were also excluded; however, the reference lists of relevant reviews were examined to identify additional potential studies for inclusion. Studies published from 2000 onwards and published in English were eligible for inclusion.

Studies reporting a KT strategy aimed at supporting or improving health systems or policy decisions to support SRMNCAH and well-being were eligible for inclusion. KT strategies aimed at patients, caregivers, healthcare providers, healthcare management, health systems, policymakers, civil society organisations and funder or donors, within or outside the health sector, were also eligible as long as it was in the context of SRMNCAH and well-being. Studies that targeted these stakeholders outside of SRMNCAH and well-being were excluded. KT strategies addressing HIV were excluded, unless specifically focused on SRMNCAH. Study outcomes relating to the effectiveness or implementation of the KT strategy or activity and SRMNCAH were included. Studies that did not report primary outcomes relevant to KT or SRMNCAH were excluded.

### Search strategy and information sources

A comprehensive search strategy was developed with support from an experienced library scientist. The search strategy was peer-reviewed by a second library scientist using the Peer Review of Electronic Search Strategy guidelines to ensure a comprehensive and high-quality search strategy was developed.[17] An electronic database search of MEDLINE ALL (Ovid), Embase (Elsevier Embase.com), CINAHL with Full Text (EBSCOhost) and Web of Science (SCI-EXPANDED, SSCI, A&HCI, CPCI-S, CPCI-SSH, ESCI; Clarivate) was executed on 29 May 2020, and results were limited from January 2000 to the search date (see online supplemental file 1). No search filters or other limits were applied. Search strategy citations were imported into Covidence, an online systematic review management software, and duplicates were removed automatically in Covidence prior to screening.[18] Reference lists of all included studies, as well as those of any relevant systematic reviews, were screened by one reviewer and verified by a second. Additionally, a search of grey literature was undertaken by a reviewer in July 2020. Search terms were applied in Google and relevant

website links were clicked through to identify any reports or literature. The reviewer clicked through each relevant website and used reference chaining within the website to ensure any and all relevant literature was identified. Google results were browsed until the reviewer went two pages (20 results) without clicking on a potentially relevant result. The website URL links were compiled and verified for inclusion by another team member.

### Selection of sources of evidence

Reviewers independently screened titles and abstracts against the inclusion criteria, and all conflicts were resolved by a third reviewer. Full-text articles were then reviewed and assessed against the inclusion criteria by one reviewer then verified by the second reviewer. Uncertainties at this stage were resolved through discussions with the research team. The reference lists of included full-text articles were then reviewed to identify other potential studies for inclusion.

### Data charting and data items

Data were extracted and mapped to four categories: descriptive details of the study (eg, authors and year, country, sample characteristics, study design, decision-making level, and SRMNCAH priority), characteristics of the KT strategy or activity, (eg, description of individual components, mode of delivery, stakeholder involvement), outcomes and direction of effect, and barriers and enablers identified by study authors. Study data were extracted using the data extraction tool by one reviewer and was verified by another to ensure all relevant data were captured. Critical appraisal of individual sources of evidence were not conducted.

### Synthesis of results

Following data extraction, the income level of each country identified from the studies was determined using the World Bank classification.[19] Study details were grouped into four categories that aligned with each of the research questions: KT strategy, stakeholder engagement, reported outcomes and types of barriers and enablers identified. Quantitative summaries and thematic analysis were then applied to each grouping to identify potential trends across country income levels and SRMNCAH thematic areas. Study data were also mapped on to the PMNCH's high-level outcomes of interest identified in the 2021–2025 Strategy (eg, policy, service delivery, financing)[10] and narrative summaries were produced.

To facilitate summarising content associated with the KT strategies, details of each KT strategy were mapped to the Behaviour Change Wheel (BCW).[20] The BCW provides a synthesis of 19 behaviour change theories in a comprehensive, theory-based tool that can be used to identify important behaviour change elements to consider in intervention design.[21] The BCW includes nine intervention functions that can be used to guide intervention content and design. It also includes seven policy categories to guide implementation of behaviour change interventions and policies.[20] KT strategies were mapped to relevant BCW intervention functions and policy categories by two independent researchers. Researchers met to review BCW coding and discrepancies were resolved through discussion.

### Patient and public involvement

This scoping review was conducted without the involvement of patients or members of the public.

## RESULTS

### Selection of sources of evidence

The search strategy returned 11 190 studies for screening. After removing 3626 duplicates, 7564 titles and abstracts were screened by reviewers. This stage identified 212 full-text studies to review. Following full-text analysis, 154 studies were excluded, resulting in 58 studies included in the review. A review of the reference lists of included studies identified 26 additional studies that met the inclusion criteria. The grey literature search identified one study for inclusion; five additional studies were included from screening the reference list of a relevant systematic review. This resulted in a final total of 90 included studies in the scoping review. The selection process and sources of evidence are summarised in a PRISMA-ScR flow diagram (figure 1).[22]

### Characteristics of sources of evidence

A summary of the characteristics of the 90 included studies can be found in table 1, with a comprehensive description of studies found in online supplemental file 2. All studies were published between 2000 and 2020, with an increase in relevant publications since 2006. Just over a third (34%) were published between 2011 and 2015[23–53] and another 34% between 2016 and 2020.[54–84] Thirty-two per cent of studies were quasi-experimental designs,[23 25 26 28 31 32 34 35 44 50 51 59–61 67 68 75 76 83 85–92] with observational (22%),[36 37 39 43 45 52 71 72 77 80 93–102] mixed-methods (17%),[40–42 47 53 56–58 64 69 82 103–106] experimental (16%)[24 27 38 62 66 74 78 79 107–112] and qualitative (11%)[29 30 33 49 54 55 63 70 81 113] designs also identified. Two editorials were also included.[65 73]

### Country income levels

The majority of the studies (52%) were conducted in high-income countries,[23–25 27–29 31 33–35 38 40 41 43–45 49 53 56 57 62 66–68 71 73 74 74 80 83 87 93–95 97 98 100 102–105 107 108 108 109 112] including Canada, the USA and Australia. Middle-income countries were the setting for 36% studies,[26 30 32 36 39 42 46–48 50 52 63–65 75–77 81 82 85 86 88–92 96 101 110 111 113] including Nepal, Egypt and Zambia, with most middle-income countries considered low-middle income (81%) and 19% considered upper-middle income. Only 4% of included studies were located in low-income countries, such as Uganda and Madagascar.[58–60 99] Seven countries (ie, Bangladesh, Cameroon, Ethiopia, Myanmar, Nigeria, Solomon Islands and Uganda) across 14% of studies represented humanitarian or highly fragile settings,[37 42 54 58 59 61 70 72 85 99] according

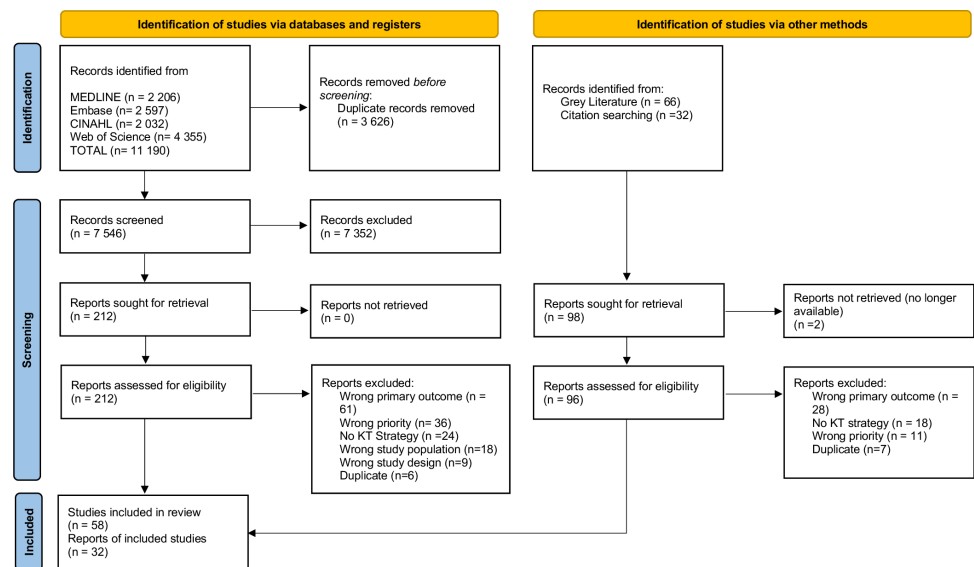

**Figure 1** Preferred Reporting Items for Systematic Reviews and Meta-Analyses-Extension for Scoping Reviews flow diagram of literature search. KT, knowledge translation.

to Organisation for Economic Co-operation and Development.[114] Eight per cent of studies included multi-income countries, with six targeting both low-income and middle-income countries (eg, Uganda and Peru),[37 54 61 69 70 72] and one targeting middle-income and high-income countries (eg, Brazil and Chile).[106] As results were reported by country in these seven studies, we analysed these studies at each of the appropriate income levels.

### SRMNCAH priorities
Child health and well-being was the most commonly identified priority, with 30% targeting these concerns.[27 29 32 33]

| Table 1 | Overview of characteristics of included studies (n=90) | | | | | |
|---|---|---|---|---|---|---|
| **Publication date** | **Country by income level** | **Study design** | **SRMNCAH priorities** | **Examples of SRMNCAH health topics** | **Settings** | **PMNCH function of interest** |
| **2000–2005:** n=8 (9%)<br>**2006–2010:** n=20 (22%)<br>**2011–2015:** n=31 (34%)<br>**2016–2020:** n=31 (34%) | **Low:** n=4 (4%)<br>**Low and middle:** n=6 (7%)<br>**Lower middle:** n=26 (81%)<br>**Upper middle:** n=6 (19%)<br>**Middle and High:** n=1 (1%)<br>**High:** n=47 (52%) | **Experimental:** n=14 (16%)<br>**Quasi-experimental:** n=29 (32%)<br>**Observational:** n=20 (22%)<br>**Qualitative:** n=10 (11%)<br>**Mixed methods:** n=15 (17%)<br>**Editorials:** n=2 (2%) | **Adolescent:** n=4 (4%)<br>**Child:** n=28 (31%)<br>**Maternal:** n=14 (16%)<br>**Newborn or stillbirths:** n=20 (22%)<br>**Maternal and newborn:** n=8 (9%)<br>**Maternal and child:** n=2 (2%)<br>**Maternal and child and newborn:** n=2 (2%)<br>**SRHR:** n=12 (13%) | **Adolescent:** substance use[107]<br>**Child:** Child nutrition[27 35 38 67 74 78 79]<br>**Maternal:** postpartum depression,[45] eclampsia and pre-eclampsia[113]<br>**Newborn or stillbirths:** Newborn sleep,[23] newborn vaccination[97]<br>**SRHR:** Family planning[42 49 65 69 70 101] | **Hospitals:** n=35 (39%)<br>**Community:** n=23 (25%)<br>**Childcare centres or schools:** n=10 (11%)<br>**Primary Care:** n=7 (8%)<br>**Government departments:** n=3 (3%) | **Financing:** n=0<br>**Policymaking:** n=17 (19%)<br>**Service delivery:** n=73 (81%) |

PMNCH, Partnership for Maternal, Newborn and Child Health; SRHR, sexual and reproductive health and rights; SRMNCAH, sexual, reproductive, maternal, newborn, child and adolescent health.

[35 36 38 40 43 46 47 51 57 62 67 74 78–80 83 85 89 91 94 98–100 105 108 110] Twenty-eight per cent addressed newborn health and well-being or stillbirths,[23 25 26 28 31 34 50 59 60 60 64 68 81 82 86 88 90 95–97 109 111] and a quarter of studies addressed maternal health and well-being.[24 37 39 44 45 52 54 55 58 63 72 73 75–77 87 92 93 103 104 112 113] Studies addressing sexual and reproductive health and rights (SRHR (12%))[30 38 42 61 65 66 69–71 101 106] and adolescent health and well-being (4%)[33 41 56 107] priorities were identified, but in a lower volume compared with studies addressing maternal, child and newborn or stillbirth priorities. Some studies (13%) also included multiple SRMNCAH priorities, with eight targeting both maternal and newborn health and well-being,[37 46 54 81 82 84 90 92] two targeting maternal and child health and well-being[52 72] and two targeting maternal, child and newborn health and well-being.[75 76] These studies were analysed at each of the appropriate priority levels.

### PMNCH high-level Outcomes (2021–2025)

Across the priority high-level outcomes of the PMNCH 2021–2025 Strategy, the majority of included studies (81%) addressed *service delivery*. *Policymaking* was also commonly identified (19%). None of the included KT strategies addressed *financing* as a central focus.

### Settings

KT strategies were primarily implemented in hospitals (39%), community (25%) or primary care (8%) settings. Three per cent of strategies were conducted within government health departments or agencies.[71 73 108] There were rarely enough details provided to explore specific departmental settings in hospitals, although four hospital studies were conducted in newborn and paediatric intensive care units,[26 31 89 109] one in a labour and delivery department[93] and one specified the hospital was in a rural setting.[86] Community settings were also described in limited detail; however, eight studies were conducted in childcare centres,[27 35 38 62 67 74 79 83] two in schools[78 107] and one in a mental health community clinic.[56]

### Synthesis of KT strategies and activities
### BCW intervention functions

Mapping the KT strategies to the BCW intervention functions identified that all but one study contained at least one intervention function. All nine BCW intervention functions were identified across the KT strategies, with *Education*, *Training* and *Environmental Restructuring* the most common (table 2).

*Education* was the most commonly identified element from the included studies, identified in 81% of studies. Types of *Education* provided included modules, staff workshops, slides and other resources. This function was identified in 73 KT strategies across country income levels (88% in high-income countries, 80% of low-income countries and 77% of middle-income countries). At the SRMNCAH priority level, 91% of child health and well-being strategy included an *Education* function. *Education* was also identified in 81% of the maternal health

interventions, as well as 75% of each of the SRHR, and adolescent and well-being interventions.

*Training* was identified in 51% of strategies. *Training* typically included conveying skills to staff members and healthcare providers. This element was included in 60% of the KT strategies aimed at low-income countries, 52% of high-income countries and 41% of middle-income countries. Strategies stratified by SRMNCAH priority identified *Training* was utilised consistently across each priority.

*Environmental Restructuring* was identified in 50% of the KT strategies. This element included reorganising how services were provided, how health centres were set up and adding additional resources (eg, tools, team members) in the health centre or health system to facilitate use of the KT strategy. This element was identified at all three income levels but was more likely to be used in high-income countries (58%), compared with middle (41%) and low-income (40%) countries. Sixty-nine per cent of child health and well-being strategies included this function, as well as 50% of the SRHR strategies. *Environmental Restructuring* was only applied in one (25%) of the four adolescent health and well-being strategies.

### BCW policy categories

Sixty per cent of studies included policy category content, with six of the seven BCW policy categories were identified across the strategies (table 2). No strategies applied content related to the *Fiscal* category option. *Guidelines* and *Service Provision* were the most commonly identified, with G*uidelines* identified in 28% of KT strategies and *Service Provision* identified in 20% of studies. *Guidelines* were implemented in 50% of the low-income countries, compared with only 31% of middle-income and 25% of the high-income countries. *Guidelines* were most frequently used in 50% of KT strategies related to sexual or reproductive health and rights, and 36% of maternal health and well-being strategies.

*Service Provision* was implemented in 89% of the high-income countries, compared with only 5% of middle-income countries and none of the low-income countries. This policy category was divided across SRMNCAH priorities. SRHR included *Service Provision* in 25% of strategies, and 25% of strategies targeting adolescent health and well-being included this category.

### Mode of KT delivery

Seventy-two per cent of KT strategies identified the mode of KT delivery, with the majority using one mode of delivery, while 27% used two or more modes. The majority (59%) of these studies used in-person delivery as the sole mode of delivery. Of the multimodal delivery strategies, in-person delivery was also included as a mode in all but one study.[111] High-income countries used more multimode interventions, such as in-person with additional online or web-based components, compared with those in low-income and middle-income countries. Studies published between 2018 and 2020 have used

**Table 2** Summary of BCW intervention functions and policy categories identified in KT strategies by country income level and SRMNCAH priority

**BCW intervention functions**

| Intervention function and definition[20] | Intervention function stratified by country income level* | Intervention function stratified by SRMNCAH priority† |
|---|---|---|
| **Education (n=73; 81%)** (Increasing knowledge or understanding) | Low: n=8 (80%)<br>Middle: n=30 (77%)<br>High: n=45 (88%) | Adolescent: n=3 (75%)<br>Child: n=29 (91%)<br>Maternal: n=21 (78%)<br>Newborn or stillbirths: n=23 (77%)<br>SRHR: n=9 (75%) |
| **Training (n=46; 51%)** (Imparting skills) | Low: n=6 (60%)<br>Middle: n=19 (41%)<br>High: n=26 (52%) | Adolescent: n=2 (50%)<br>Child: n=14 (44%)<br>Maternal: n=12 (46%)<br>Newborn or stillbirths: n=16 (53%)<br>SRHR: n=7 (58%) |
| **Environmental restructuring (n=45; 50%)** (changing the physical or social context) | Low: n=4 (40%)<br>Middle: n=16 (41%)<br>High: n=30 (58%) | Adolescent: n=1 (25%)<br>Child: n=22 (69%)<br>Maternal: n=10 (38%)<br>Newborn or stillbirths: n=14 (47%)<br>SRHR: n=6 (50%) |
| **Enablement (n=22; 24%)** (increasing means/reducing barriers) | Low: n=5 (50%)<br>Middle: n=6 (15%)<br>High: n=14 (29%) | Adolescent: n=1 (25%)<br>Child: n=8 (25%)<br>Maternal: n=6 (23%)<br>Newborn or stillbirths: n=4 (13%)<br>SRHR: n=2 (13%) |
| **Persuasion (n=11; 12%)** (communication used to induce positive or negative feelings or stimulate action) | Low: n=1 (10%)<br>Middle: n=2 (5%)<br>High: n=10 (21%) | Adolescent: n=0<br>Child: n=8 (25%)<br>Maternal: n=3 (12%)<br>Newborn or stillbirths: n=1 (3%)<br>SRHR: n=1 (8%) |
| **Modelling (n=7; 8%)** (providing an example for people to aspire to or imitate) | Low: n=0<br>Middle: n=3 (8%)<br>High: n=4 (8%) | Adolescent: n=0<br>Child: n=2 (6%)<br>Maternal: n=2 (8%)<br>Newborn or stillbirths: n=2 (7%)<br>SRHR: n=1 (8%) |
| **Incentivisation (n=5; 6%)** (creating expectation of reward) | Low: n=0<br>Middle: n=1 (3%)<br>High: n=4 (8%) | Adolescent: n=0<br>Child: n=4 (13%)<br>Maternal: n=0<br>Newborn or stillbirths: n=0<br>SRHR: n=1 (8%) |
| **Coercion (n=2; 2%)** (creating expectation of punishment or cost) | Low: n=1 (10%)<br>Middle: n=1 (3%)<br>High: n=0 | Adolescent: n=0<br>Child: n=0<br>Maternal: n=1 (4%)<br>Newborn or stillbirths: n=2 (7%)<br>SRHR: n=0 |
| **Restriction (n=1; 1%)** (using rules to reduce the opportunity to engage in the target behaviour) | Low: n=0<br>Middle: n=1 (3%)<br>High: n=0 | Adolescent: n=0<br>Child: n=0<br>Maternal: n=0<br>Newborn or stillbirths: n=1 (3%)<br>SRHR: n=0 |

**BCW policy categories**

| Policy category and definition [20] | Policy category stratified by country income level* | Policy category stratified by SRMNCAH priority† |
|---|---|---|
| **Guidelines (n=25; 28%)** (creating documents that recommend or mandate practice) | Low: n=5 (50%)<br>Middle: n=12 (31%)<br>High: n=12 (25%) | Adolescent: n=1 (25%)<br>Child: n=6 (19%)<br>Maternal: n=9 (35%)<br>Newborn or stillbirths: n=8 (27%)<br>SRHR: n=5 (42%) |

Continued

**Table 2** Continued

**BCW intervention functions**

| Intervention function and definition[20] | Intervention function stratified by country income level* | Intervention function stratified by SRMNCAH priority† |
|---|---|---|
| **Service Provision (n=18; 20%)** (delivering a service) | Low: n=0 Middle: n=2 (5%) High: n=15 (31%) | Adolescent: n=1 (25%) Child: n=6 (19%) Maternal: n=4 (15%) Newborn or stillbirths: n=7 (23%) SRHR: n=3 (25%) |
| **Communication/marketing (n=12; 13%)** (using print, electronic, telephonic or broadcast media) | Low: n=2 (20%) Middle: n=4 (10%) High: n=8 (17%) | Adolescent: n=0 Child: n=5 (16%) Maternal: n=3 (12%) Newborn or stillbirths: n= (13%) SRHR: n=3 (25%) |
| **Environmental/social planning (n=11; 12%)** (designing and/or controlling the physical or social environment) | Low: n=1 (10%) Middle: n=4 (10%) High: n=6 (12%) | Adolescent: n=0 Child: n=4 (13%) Maternal: n=5 (19%) Newborn or stillbirths: n=5 (17%) SRHR: n=0 |
| **Regulation (n=7; 8%)** (establishing rules or principles of behaviour or practice) | Low: n=1 (10%) Middle: n=2 (5%) High: n=5 (10%) | Adolescent: n=0 Child: n=3 (9%) Maternal: n=1 (3%) Newborn or stillbirths: n=4 (13%) SRHR: n=0 |
| **Legislation (n=3; 3%)** (making or changing laws) | Low: n=0 Middle: n=3 (8%) High: n=0 | Adolescent: n=0 Child: n=0 Maternal: n=1 (4%) Newborn or stillbirths: n=0 SRHR: n=2 (17%) |

*Please note country income levels include seven multicountry studies (n=97).
†Please note priorities include 12 multipriority studies (n=104).
BCW, Behaviour Change Wheel; SRHR, sexual and reproductive health and rights.

evolving trends in technology, such as webinars and social media,[73 83] as modes of delivery.

### Synthesis of stakeholder involvement

There was an overall lack of description provided on how stakeholders were involved in designing or implementing KT strategies, with 31% of studies not providing any description of stakeholder engagement. Of the remaining studies that did provide details, the level of detail varied by article, with some simply acknowledging stakeholders were engaged, while others provided a more comprehensive view of the stakeholder groups involved and their roles. Commonly identified stakeholder groups included: government and policymakers, healthcare providers, civil society organisations, members of the public and members of the research community (table 3). Engagement with these groups was distributed across country income level, with two notable exceptions. First, civil society organisations were more likely to be engaged in low (60%) and middle-income (48%) countries compared with high-income (27%) countries. Additionally, government and policymakers were engaged by low (50%) and middle-income (62%) countries much more often than in high-income (8%) countries.

Stakeholder engagement was dispersed across the SRMNCAH priorities. Priorities addressing SRHR were more likely to include policymakers and government as well as civil society organisations compared with other priorities. Half of strategies addressing adolescent health and well-being, while 25% of SRHR engaged healthcare providers including clinicians, nurses and allied healthcare professionals. Involvement of researcher communities was identified across all six priorities but was rarely used in newborn health and well-being or stillbirth strategies (7%).

### Synthesis of outcomes

Nearly 80% of KT strategy outcomes were reported at only a single outcome level (eg, patient or healthcare provider), with 20% studies reporting multiple outcome levels (eg, healthcare provider and system). At the single outcome level, 38% of outcomes were measured at the healthcare provider level (eg, increased knowledge) and 29% at system level (eg, reductions in safety incidents). Only 5% of strategies included patient-level outcomes (eg, improved newborn sleep). Patient outcomes were more likely to be included in multilevel outcomes, along with healthcare provider and system level (eg, immunisation

Table 3  Summary of stakeholder engagement by country income level and SRMNCAH priority

| Type of stakeholders involved in KT strategy | Stakeholders by country income level* | Stakeholders by SRMNCAH priority† |
|---|---|---|
| Policymakers and government | Low: n=5 (50%) Middle: n=24 (62%) High: n=4 (8%) | Adolescent: n=1 (25%) Child: n=4 (13%) Maternal: n=9 (35%) Newborn or stillbirths: n=3 (10%) SRHR: n=7 (63%) |
| Healthcare providers and administrators | Low: n=5 (50%) Middle: n=17 (44%) High: n=23 (48%) | Adolescent: n=2 (50%) Child: n=10 (31%) Maternal: n=11 (42%) Newborn or stillbirths: n=9 (30%) SRHR: n=3 (25%) |
| Civil society organisations | Low: n=6 (60%) Middle: n=19 (48%) High: n=13 (27%) | Adolescent: n=1 (25%) Child: n=4 (13%) Maternal: n=3 (13%) Newborn or stillbirths: n=2 (7%) SRHR: n=7 (63%) |
| Public | Low: n=2 (20%) Middle: n=1 (3%) High: n=10 (21%) | Adolescent: n=2 (50%) Child: n=3 (9%) Maternal: n=1 (4%) Newborn or stillbirths: n=2 (7%) SRHR: n=2 (17%) |
| Research community | Low: n=3 (30%) Middle: n=10 (26%) High: n=17 (35%) | Adolescent: n=1 (25%) Child: n=9 (28%) Maternal: n=10 (38%) Newborn or stillbirths: n=2 (7%) SRHR: n=3 (25%) |

*Please note country income levels include seven multicountry studies (n=97).
†Please note priorities include 12 multipriority studies (n=104).
KT, knowledge translation; SRHR, sexual and reproductive health and rights; SRMNCAH, sexual, reproductive, maternal, newborn, child and adolescent health.

rates, quality of care provided and healthcare system use) outcomes. Of the multilevel outcome studies, 17% had outcomes at two levels (eg, patient and healthcare providers, healthcare providers and system or patient and system) and 4% of studies included three-level outcomes (eg, patient, healthcare provider and system).

Healthcare provider outcomes were mostly reported in high-income (44%) and low-income (40%) countries, compared with 26% of middle-income countries, while system-level outcomes were more common in low-income (40%) and middle-income (41%) countries. Healthcare

provider outcomes were identified across the SRMNCAH priorities, ranging from 23% to 75% for maternal health and well-being strategies to those for adolescent well-being, respectively. While fewer maternal health and well-being strategies included healthcare provider-level outcomes, nearly half of these strategies (48%) were aimed at system-level outcomes. Childhood health and well-being strategies were also most likely to address multi-level outcomes compared with all other priorities.

### Synthesis of barriers and enablers

Few studies reported barriers and enablers to using KT strategies to promote the use of evidence in SRMNCAH and well-being. Fewer than half of the studies (43%) outlined barriers and even fewer identified enablers (40%) to their KT strategies. When studies included barriers and enablers, the level of detail provided varied across the studies, with some studies providing a brief list of these factors while other studies provided more detailed descriptions and how each affected the KT strategy. A summary of identified barriers and enablers can be found in table 4.

### Identified barriers

Limited resources was the most commonly reported barrier for using KT strategies across countries of all income levels and SRMNCAH priorities. In 56% of strategies, limited resources referred to physical (eg, funding) or human resources (eg, healthcare staff) constraints. Low-income countries were more likely to report limited resources (40%) compared with high-income countries (19%). In terms of SRMNCAH priority, 50% of SRHR strategies identified limited resources as barriers to using KT strategies.

Second, time constraints were reported in 21% of high-income countries. This type of barrier delayed the implementation process, including fidelity of KT strategies. Lastly, negative attitudes were the third commonly reported barrier, reported by 30% of low-income countries. Examples of negative attitudes included resistance to change, lack of confidence and poor 'buy-in' for using KT strategies.

### Identified Enablers

Supportive stakeholder involvement was the most commonly identified enabler to KT strategies. Two-thirds of studies which reported enablers identified the importance of developing supportive relationships with stakeholders, and that the partnerships forged supported the implementation of the KT strategy. Supportive stakeholder involvement included successful collaboration and partnerships with, but not limited to, healthcare providers, government bodies or non-profit organisations. This enabler was identified across all country income levels but was most common in low-income countries (40%). In terms of SRMNCAH priority, 67% of SRHR studies reported supportive stakeholder involvement as enablers.

**Table 4** Summary of barriers and enablers identified across strategies by country income level and SRMNCAH priority

Barriers identified across studies (n=39)

| | Stratified by country income level* | Stratified by SRMNCAH priority† |
|---|---|---|
| Limited resources (n=22) | Low: n=4 (40%)<br>Middle: n=9 (23%)<br>High: n=9 (19%) | Adolescent: n=2 (50%)<br>Child: n=6 (19%)<br>Maternal: n=6 (23%)<br>Newborn or stillbirths: n=2 (7%)<br>SRHR: n=6 (50%) |
| Time constraints (n=12) | Low: n=0<br>Middle: n=2 (5%)<br>High: n=10 (21%) | Adolescent: n=1 (25%)<br>Child: n=5 (16%)<br>Maternal: n=2 (8%)<br>Newborn or stillbirths: n=2 (7%)<br>SRHR: n=2 (13%) |
| Negative attitudes (n=10) | Low: n=3 (30%)<br>Middle: n=3 (8%)<br>High: n=5 (10%) | Adolescent: n=1 (25%)<br>Child: n=2 (6%)<br>Maternal: n=3 (12%)<br>Newborn or stillbirths: n=1 (3%)<br>SRHR: n=3 (25%) |
| Lack of knowledge (n=9) | Low: n=1 (10%)<br>Middle: n=3 (8%)<br>High: n=5 (10%) | Adolescent: n=1 (25%)<br>Child: n=2 (6%)<br>Maternal: n=4 (15%)<br>Newborn or stillbirths: n=3 (10%)<br>SRHR: n=1 (8%) |
| Lack of training (n=7) | Low: n=0<br>Middle: n=0<br>High: n=7 (15%) | Adolescent: n=1 (25%)<br>Child: n=4 (13%)<br>Maternal: n=2 (8%)<br>Newborn or stillbirths: n=0<br>SRHR: n=0 |
| Poor engagement with stakeholders (n=7) | Low: n=0<br>Middle: n=4<br>High: n=3 | Adolescent: n=1 (25%)<br>Child: n=3 (9%)<br>Maternal: n=3 (12%)<br>Newborn or stillbirths: n=1 (3%)<br>SRHR: n=1 (8%) |
| **Enablers identified across studies (n=36)** | | |
| | Stratified Country Income Level* | Stratified by SRMNCAH Priority** |
| Supportive stakeholder involvement (n=24) | Low: n=4 (40%)<br>Middle: n=10 (26%)<br>High: n=10 (21%) | Adolescent: n=2 (50%)<br>Child: n=3 (9%)<br>Maternal: n=7 (27%)<br>Newborn or stillbirths: n=4 (13%)<br>SRHR: n=8 (67%) |
| Access to resources (n=8) | Low: n=2 (20%)<br>Middle: n=2 (5%)<br>High: n=4 (8%) | Adolescent: n=0<br>Child: n=1 (3%)<br>Maternal: n=3 (12%)<br>Newborn or stillbirths: n=3 (10%)<br>SRHR: n=2 (17%) |
| Access to knowledge (n=8) | Low: n=4 (40%)<br>Middle: n=4 (10%)<br>High: n=0 | Adolescent: n=0<br>Child: n=2 (6%)<br>Maternal: n=3 (12%)<br>Newborn or stillbirths: n=2 (7%)<br>SRHR: n=3 (25%) |
| Positive attitudes or empowerment (n=7) | Low: n=2 (20%)<br>Middle: n=3 (8%)<br>High: n=2 (4%) | Adolescent: n=0<br>Child: n=2 (6%)<br>Maternal: n=4 (15%)<br>Newborn or stillbirths: n=1 (3%)<br>SRHR: n=0 |

Continued

**Table 4** Continued

Barriers identified across studies (n=39)

|  | Stratified by country income level* | Stratified by SRMNCAH priority† |
|---|---|---|
| Skills and training (n=6) | Low: n=0<br>Middle: n=0<br>High: n=6 (13%) | Adolescent: n=0<br>Child: n=3 (9%)<br>Maternal: n=2 (8%)<br>Newborn or stillbirths: n=0<br>SRHR: n=1 (8%) |

*Please note country income levels include seven multicountry studies (n=97).
†Please note priorities include 12 multipriority studies (n=104).
SRHR, sexual and reproductive health and rights; SRMNCAH, sexual, reproductive, maternal, newborn, child and adolescent health.

Second, access to resources and knowledge was the next commonly reported enablers for using KT strategies. Resources in these articles included financial as well as human resources. This enabler was more common in low-income countries (20%) compared with middle-income (8%) or high-income (5%) countries. This enabler was identified across the SRMNCAH priorities, with the exception of adolescents' health and well-being. Finally, only high-income countries reported skills and training as enablers for using KT strategies.

## DISCUSSION

This rapid scoping review identified 90 studies published since 2000 which utilised KT strategies and policies to support the use of evidence for improving SRMNCAH and well-being. While a wide range of studies across country income levels and SRMNCAH priorities were identified, most KT strategies were implemented in high-income countries and focused on maternal, newborn or stillbirths, or child health and well-being topics. The review identified key gaps in KT interventions to support evidence-based decision-making for SRHR, adolescent health and well-being and SRMNCAH financing. Our findings illustrate the majority of KT strategies included an *education* component and strategies were commonly aimed at addressing healthcare provider and system-level outcomes. Across PMNCH outcomes of interest, most strategies address *service delivery* or *policymaking*, with none addressing SRMNCAH financing. While few details were typically provided on stakeholder engagement, or on barriers and enablers in the KT process, it was noted that collaboration with stakeholders and building partnerships with local actors, such as government or health authorities, facilitated use of KT strategies.

Low-income and middle-income countries identified a lack of resources (eg, funding, staff, physical resources) as the most common barrier to implementing and sustaining KT strategies. While lack of resources were also identified in high-income countries, these barriers look different based on country income level and are more often pronounced in low-income and middle-income countries.[113] As we only identified 10 KT strategies implemented or evaluated in low-income countries, this may

suggest more work is needed in these countries to move evidence into practice to improve SRMNCAH outcomes, while addressing the economic, resource and health system barriers they experience.[115] Using a tailored approach to specifically address the unique barriers in low-income countries may help support the successful implementation of KT strategies and improve maternal, child and adolescent health outcomes within these countries.[115]

Engaging with stakeholders was viewed as a key enabler to KT strategies included in this review. Among the groups of stakeholders involved, low-income and middle-income countries were more likely to include members of civil societies and non-governmental organisations compared with high-income countries. Additionally, government and policymakers were engaged by low-income and middle-income countries much more often than in high-income countries. Involving civil societies in KT, especially among low-income countries, is crucial as these organisations often have the capacity and resources required to implement recommendations.[116] Drawing on civil society stakeholder groups throughout the design and implementation of KT strategies may provide vital support to help facilitate the implementation and evaluation of KT strategies.

### SRMNCAH and well-being

Maternal health and well-being strategies addressed preterm labour management, labour and delivery outcomes and perinatal care, although primarily in high-income countries.[63 84 102 103 112] These strategies typically included utilising clinical guidelines and education for nurses to help support improvements in these maternal outcomes. Our review identified a critical gap around effective KT approaches to support evidence-based SRMNCAH interventions in low-income countries. Yet, these women and communities experience more barriers to high-quality healthcare, including trained healthcare providers during childbirth.[117 118] More investments and capacity strengthening efforts are needed to support KT interventions in low-income settings, thus advancing evidence-based policy and practice in settings where the needs are the most dire. This includes humanitarian and fragile settings, which bear a disproportionate burden

of poor SRMNCAH outcomes. For instance, maternal mortality increases by 11% on average in conflict zones and by 28% in the worst-hit areas.[119] Additionally, more than 10 million deaths in children younger than 5 years can be attributed to conflict between 1995 and 2015 globally.[119] In addition, up to a third of girls living in a humanitarian setting report that their first sexual encounter was forced.[120]

Many of the identified strategies were aimed at improving newborn, childhood and maternal mortality rates, particularly in low-income and middle-income countries.[23 37 52 59 72 81 86 90] Strategies included education sessions targeting healthcare providers to ensure quality care is provided to support health outcomes among these groups, and one strategy outlined an advocacy campaign targeting government members to support the provision of free health services for pregnant women and children to help reduce mortality rates.[52] Providing educational and training opportunities to healthcare providers is vital to targeting health system barriers often experienced in low-income and middle-income countries.[118 121] However, it is recommended to include stakeholder groups within and beyond the health system in these strategies, and these multisectoral stakeholder strategies will need to be sustainable to ensure continuous access to high-quality healthcare for children.

Our review identified an overall lack of relevant KT strategies addressing adolescent health and well-being. This is a significant gap in the literature as adolescents represent over 1.2 billion of today's population, and 90% of these youth living in low-income and middle-income countries.[122 123] While the adolescent studies included in this review addressed issues including substance use, mental health and overall health needs,[41 53 56 107] there is a lack of research into supporting dietary and lifestyle choices and SRHR in this population. Focusing on developing KT programmes and policies to address the health needs of adolescents is essential to supporting the transition into healthy adulthood. Future KT programmes and policies should be codesigned with adolescents and youth to harness their capacity and advocacy skills and to ensure their unique health needs are being addressed.[123]

### SRMNCAH financing
Our review identified a key gap in effective KT interventions to support evidence-based decision-making on SRMNCAH financing. This finding highlights the need for greater efforts to ensure that robust health financing evidence is used to strengthen public financial management systems pertaining to SRMNCAH.[124] This is critical to support efficient spending in times of COVID-19, whereby governments are struggling with shrinking fiscal spaces and major disruptions to essential SRMNCAH services.[125]

### Implications for future KT strategies
Findings from this scoping review identified educational interventions aimed at changing healthcare provider behaviours to improve provision of care are implemented across all countries and SRMNCAH priorities. Future work should build on these good practices to address issues around SRHR, adolescent health and well-being, and SRMNCAH financing. It will also be essential for teams designing and implementing KT strategies to integrate stakeholder groups early on in the process and codesign KT interventions to optimise the success of work.[126] Depending on the complexity and scalability of the KT strategies, especially in low-income and middle-income countries, there is an opportunity to identify and address barriers to optimal implementation. Utilising models, such as the Ottawa Model for Research Use recommended by Santesso and Tugwell, throughout the KT process may prove useful to support effective implementation of KT strategies and positive outcomes.[115 127]

Across the studies identified in our review, only 16% utilised an experimental design, with most studies being observational in nature, thus impeding the assessment of the effectiveness of KT interventions. Future research could benefit from more experimental study designs (eg, cluster randomised control trials, interrupted time series, controlled before and after studies)—and eventually a systematic review and meta-analysis—to evaluate the effectiveness of SRMNCAH strategies, which could provide useful direction and guidance for KT decision-makers and policymakers.

### Limitations
Due to the rapid nature of this scoping review, it is possible we may have missed relevant KT strategies in the search. Another potential limitation is only identifying one piece of grey literature in the scoping review, which may have been due to our search strategy and strict inclusion criteria. However, our comprehensive literature and grey literature searches identified a broad range of relevant strategies across countries and SRMNCAH priorities. The challenges of identifying relevant KT strategies and policies may also be due to how varied the terminology is around KT (eg, KT, knowledge exchange, knowledge mobilisation) and how it is applied across different countries and health areas.[128] Additionally, while few KT strategies included in the review were implemented and evaluated in low-income countries, this may not reflect all of the KT strategies being implementing across these settings. Although we conducted a search of the grey literature, it is possible publication bias may have impacted our ability to include all unpublished KT work conducted to address SRMNCAH priorities in different settings.[129] Finally, findings from the review are up to date as of July 2020 and it is possible additional relevant strategies may have been published between running the search strategy and report writing. However, this review provides a comprehensive view of KT strategies which have been published over the past 20 years to address the health and well-being of women, children and adolescents.

## CONCLUSION

Most KT strategies included in this scoping review were implemented in high-income countries and aimed at maternal, newborn and child health and well-being. We identified a critical gap in the published literature of KT approaches in low-income countries and humanitarian and fragile settings. Meaningful engagement of stakeholders in KT was identified as a key enabler to enhance people-centred and gender-responsive SRMNCAH policy, service delivery and financing. Effective KT approaches are required to support the implementation and impact of multisectoral policies and interventions. As SRMNCAH outcomes worsen as a consequence of the COVID-19 pandemic, it is critical to support SRMNCAH interventions and protect the progress made to date. KT approaches have a key role to play to ensure that strategies to mitigate the disruptions to SRMNCAH services are effective, feasible and acceptable, while addressing the equity gap and ensuring that vulnerable communities are not left behind.

**Author affiliations**
¹School of Nursing, Dalhousie University, Halifax, Nova Scotia, Canada
²Pediatrics, IWK Health Centre, Halifax, Nova Scotia, Canada
³Faculty of Health, Dalhousie University, Halifax, Nova Scotia, Canada
⁴Department of Community Health and Epidemiology, Dalhousie University, Halifax, Nova Scotia, Canada
⁵Maritime SPOR SUPPORT Unit, Halifax, Nova Scotia, Canada
⁶Nova Scotia Health, Halifax, Nova Scotia, Canada
⁷Centre for Global Child Health, The Hospital for Sick Children, Toronto, Ontario, Canada
⁸The Partnership for Maternal, Newborn & Child Health, World Health Organization, Geneva, Switzerland

**Contributors** EVL, ZB, JAC, HW, RU, LB and LW developed the scoping review protocol. JAC, AJG, HDS, JK and EVL contributed to developing the initial draft of the manuscript. All authors provided feedback and revisions to the manuscript. As guarantor for this work, JAC accepts full responsibility for the conduct of the study and has access to the data.

**Funding** This work was funded by the SPOR Evidence Alliance in collaboration with the Partnership for Maternal, Newborn & Child Health (PMNCH) at the World Health Organization.

**Competing interests** None declared.

**Patient consent for publication** Not applicable.

**Ethics approval** This study does not involve human participants.

**Provenance and peer review** Not commissioned; externally peer reviewed.

**Data availability statement** Data sharing not applicable as no datasets generated and/or analysed for this study.

**ORCID iDs**
Janet A Curran http://orcid.org/0000-0001-9977-0467
Allyson J Gallant http://orcid.org/0000-0002-2933-7470
Julia Kontak http://orcid.org/0000-0002-9104-0678
Zulfiqar Bhutta http://orcid.org/0000-0003-0637-599X

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
