## [Reviewer comments · BMJ Open]

ARTICLE DETAILS

TITLE (PROVISIONAL)	Knowledge Translation Strategies for Policy and Action Focused on Sexual, Reproductive, Maternal, Newborn, Child & Adolescent Health and Wellbeing: A Rapid Scoping Review
AUTHORS	Curran, Janet; Gallant, Allyson; Wong, Helen; Shin, Hwayeon Danielle; Urquhart, Robin; Kontak, Julia; Wozney, Lori; Boulos, Leah; Bhutta, Zulfiqar; Langlois, Etienne V.

VERSION 1 – REVIEW

REVIEWER	Dean, Elizabeth University of British Columbia, Physical Therapy
REVIEW RETURNED	03-Jul-2021

GENERAL COMMENTS	Review of 'Effective Knowledge Translation Strategies for Policy and Action Focused on Sexual, Reproductive, Maternal, Newborn, Child & Adolescent Health and Wellbeing: A Rapid Scoping Review' Journal: BMJ Open Manuscript ID bmjopen-2021-053919 Overview This scoping review aimed to identify knowledge translation (KT) strategies focused on improving sexual, reproductive, maternal, newborn, child and adolescent health (SRMNCAH) and wellbeing. Four established electronic databases were searched from January 2020 (MEDLINE ALL, Embase, CINAHL, and Web of Science). The investigators reported that '... 4% of included 90 studies were conducted in low-income countries with the majority (52%) conducted in high-income countries. Studies primarily focused on maternal newborn or child health and wellbeing. Education (81%), including staff workshops and education modules, was the most commonly identified intervention component from the KT interventions. Low- and middle-income countries were more likely to include civil society organizations, government and policy makers as stakeholders compared to high-income countries.' Further, barriers to KT strategies reported were limited resources and time constraints. Enablers reported were supportive stakeholder participation throughout the KT process. The investigators concluded that 'Gaps exist among KT strategies and there is '...a need to support stakeholder engagement in KT interventions across the continuum of SRMNCAH services, especially in low-income countries.' Finally, they state that 'Future efforts in SRMNCAH should prioritize working with multi-sectoral stakeholders to implement successful KT strategies and policies.' Substantive Comments
---

	On reading the title and early on in the manuscript, I was believing the review topic was overly ambitious, and the results would not be meaningful. However, for this preliminary study, a scoping review of the topic with an emphasis on 'KT for policy and action', I now believe the broad target was justifiable. Although clearly 'maternal and child health' is the unifying theme, the scope is 'vast'. Having said that, the methods are sound, and the investigators have partitioned the data effectively to generally tease out the use of KT strategies across the spectrum of interest. What is particularly important about this scoping review is precisely its focus on 'KT and policy and action'. In the health sciences literature, we need many more studies with this focus. KT is woefully inadequate, in my view, particularly related to policy and action to maximize people's health and wellbeing. We actually know a great deal that the general public is not getting advantage of, due to inertia of the medical establishment in effecting policy and action changes. For this, the investigators are to be congratulated. They also ground their work in Michie's Behaviour Change Wheel to provide a basis for their scoping strategy. My substantive comments are twofold. A scoping review does not typically concern itself with 'effect'. Thus, the title needs changing to: 'Knowledge Translation Strategies for Policy and Action Focused on Sexual, Reproductive, Maternal, Newborn, Child and Adolescent Health and Wellbeing: A Rapid Scoping Review' To establish 'effect' and 'success' (remove 'successful' from the end of the Abstract), a systematic review with a meta-analysis is needed, preferably of randomized clinical trials. The source studies in this scoping review would generally be classified as weak experimental designs. This needs to be addressed in a section on future studies that are indicated based on this scoping review, including the issue of 'effect'. This is just what good scoping reviews do. My second issue is one of validity. Based on the findings in terms of low- and high-income countries, the investigators imply that they reflect the 'actual' status in those countries. I believe this is a leap. Just because work may not be published, does not mean to say, KT is not happening. Having worked and practiced internationally extensively in low-, middle- and high-income countries, my impression is indeed KT in the form of education needs to improve across the board and more so in lower-income contexts, but this is true in low-income contexts in high-income countries. Again, based on my experience, some low-income countries are doing a superior job (through their generic health workers) than higher income countries with more highly trained health professionals, because ironically 'education' may be all that they have. We in high-income countries can learn a great deal from poorer African countries and rural India. I am not implying my impressions are necessarily valid or the 'truth', but they do support the need to frame the findings more conservatively and greater acknowledgement of what we do not know and need to discover in future studies. That the findings truly represent the status in low- and middle-income countries is not warranted. Rather, better to
--	--

	say 'our findings suggest that.... may...', hence, future studies are needed to validate our findings.
REVIEWER	Kele, Pierre Abomo LSTM, International Public Health
REVIEW RETURNED	12-Jul-2021
GENERAL COMMENTS	The topic is too broad and too many issues to be dealt with without a clear explanation about why wellbeing, sexual health, new born, maternal health, adolescent health, reproduction health etc. should amalgamated in one paper. This makes it difficult to follow. Different areas of the scientific literature are involved and it would be difficult to compare or to make meaningful conclusion given the scope of the paper.

VERSION 1 – AUTHOR RESPONSE

Reviewer #1 Comments	
On reading the title and early on in the manuscript, I was believing the review topic was overly ambitious, and the results would not be meaningful. However, for this preliminary study, a scoping review of the topic with an emphasis on 'KT for policy and action', I now believe the broad target was justifiable. Although clearly 'maternal and child health' is the unifying theme, the scope is 'vast'. Having said that, the methods are sound, and the investigators have partitioned the data effectively to generally tease out the use of KT strategies across the spectrum of interest. What is particularly important about this scoping review is precisely its focus on 'KT and policy and action'. In the health sciences literature, we need many more studies with this focus. KT is woefully inadequate, in my view, particularly related to policy and action to maximize people's health and wellbeing. We actually know a great deal that the general public is not getting advantage of, due to inertia of the medical establishment in effecting policy and action changes. For this, the investigators are to be congratulated. They also ground their work in Mitchie's Behaviour Change Wheel to provide a basis for their scoping strategy.	Thank you for this feedback on our scoping review. We have incorporated your suggestions and have re-named the review to remove 'Effective' from the title (see line 1).
A scoping review does not typically concern itself with 'effect'. Thus, the title needs changing to:	

'Knowledge Translation Strategies for Policy and Action Focused on Sexual, Reproductive, Maternal, Newborn, Child and Adolescent Health and Wellbeing: A Rapid Scoping Review'	
To establish 'effect' and 'success' (remove 'successful' from the end of the Abstract), a systematic review with a meta-analysis is needed, preferably of randomized clinical trials. The source studies in this scoping review would generally be classified as weak experimental designs. This needs to be addressed in a section on future studies that are indicated based on this scoping review, including the issue of 'effect'. This is just what good scoping reviews do.	We have removed 'successful' from the conclusion section of our abstract (line 78). We have also incorporated your feedback and have added details to the "Implications for Future KT Strategies" section to discuss the lack of high quality study designs included in our review. (please see lines 551-557)
My second issue of one of validity. Based on the findings in terms of low- and high-income countries, the investigators imply that they reflect the 'actual' status in those countries. I believe this is a leap. Just because work may not be published, does not mean to say, KT is not happening. Having worked and practiced internationally extensively in low-, middle- and high-income countries, my impression is indeed KT in the form of education needs to improve across the board and more so in lower-income contexts, but this is true in low-income contexts in high-income countries. Again, based on my experience, some low-income countries are doing a superior job (through their generic health workers) than higher income countries with more highly trained health professionals, because ironically 'education' may be all that they have. We in high-income countries can learn a great deal from poorer African countries and rural India. I am not implying my impressions are necessarily valid or the 'truth', but they do support the need to frame the findings more conservatively and greater acknowledgement of what we do not know and need to discover in future studies. That the findings truly represent the status in low- and middle-income countries is not warranted. Rather, better to say 'our findings suggest that.... may...', hence, future studies are needed to validate our findings.	Thank you for this feedback and we agree with your suggestions.
Reviewer #2 Comments	

The topic is too broad and too many issues to be dealt with without a clear explanation about why wellbeing, sexual health, new born, maternal health, adolescent health, reproduction health etc. should amalgamated in one paper. This makes it difficult to follow. Different areas of the scientific literature are involved and it would be difficult to compare or to make meaningful conclusion given the scope of the paper.	The topic for our scoping review is quite broad, however this is to align with the Partnership of Maternal, Newborn and Child Health (PMNCH) and their goal to advance sexual, reproductive, maternal, newborn, child and adolescents health (SRMNCAH). We have added a few lines in the introduction to introduce the PMNCH (please see lines 132-135).
---	---